# Deforestation amplifies climate change effects on warming and cloud level rise in African montane forests

Temesgen Alemayehu Abera [1,2] ✉, Janne Heiskanen [2,3], Eduardo Eiji Maeda [2,3], Mohammed Ahmed Muhammed[1,4], Netra Bhandari[1], Ville Vakkari[3,5], Binyam Tesfaw Hailu[2,4], Petri K. E. Pellikka[2,6], Andreas Hemp [7], Pieter G. van Zyl[5] & Dirk Zeuss [1]

Tropical montane forest ecosystems are pivotal for sustaining biodiversity and essential terrestrial ecosystem services, including the provision of high-quality fresh water. Nonetheless, the impact of montane deforestation and climate change on the capacity of forests to deliver ecosystem services is yet to be fully understood. In this study, we offer observational evidence demonstrating the response of air temperature and cloud base height to deforestation in African montane forests over the last two decades. Our findings reveal that approximately 18% (7.4 ± 0.5 million hectares) of Africa's montane forests were lost between 2003 and 2022. This deforestation has led to a notable increase in maximum air temperature (1.37 ± 0.58 °C) and cloud base height (236 ± 87 metres), surpassing shifts attributed solely to climate change. Our results call for urgent attention to montane deforestation, as it poses serious threats to biodiversity, water supply, and ecosystem services in the tropics.

Tropical montane forests are characterized by a unique hydroclimate and rich biodiversity, providing essential ecosystem services to humans and hosting a wide variety of endemic plants and animals[1–4]. Montane forests are the interface for the transfer of water from the atmosphere to the land, providing a stable freshwater supply to the surrounding lowlands[4,5]. Tree cover plays a key role in maintaining a sustainable water supply by intercepting additional water from clouds and fog through tree leaves and by replenishing the atmospheric water content through moisture recycling[6–9]. Tree cover cools local daytime temperatures through shading, absorption of incoming solar radiation, and evapotranspiration, which helps in maintaining a lower cloud base height for cloud water interception[8,10].

Tropical montane forests in Africa are highly threatened by deforestation and climate change, posing a serious risk to the sustainability of water supply in the region[7,11–13]. Previous studies have reported the impact of lowland deforestation on the cloud regime of Monteverde forest in Costa Rica[11,14]. An increase in cloud base height (CBH) impacts the capacity of mountain ecosystems to provide water supply by reducing the forest area under cloud cover[11], the frequency of cloud immersion, and the quantity of cloud water intercepted by trees[6,8,15]. However, the effect of montane deforestation on CBH and the sustainability of water supply in Africa remain unknown. It is unclear to what extent and magnitude the CBH has shifted in response to deforestation, and whether geographical hotspots exist that require special attention. Furthermore, whether deforestation accelerates the impacts of climate change on CBH across montane forests in Africa remains poorly understood.

[1]Department of Environmental Informatics, Faculty of Geography, Philipps-Universität Marburg, Deutschhausstraße 12, 35037 Marburg, Germany. [2]Department of Geosciences and Geography, University of Helsinki, P.O. Box 68, FI-00014 Helsinki, Finland. [3]Finnish Meteorological Institute, P.O. Box 503, FI-00101 Helsinki, Finland. [4]School of Earth Sciences, Addis Ababa University, Addis Ababa, Ethiopia. [5]Atmospheric Chemistry Research Group, Chemical Resource Beneficiation, North-West University, Potchefstroom, South Africa. [6]State Key Laboratory for Information Engineering in Surveying, Mapping and Remote Sensing, Wuhan University, Wuhan 430079, China. [7]Department of Plant Systematics, University of Bayreuth, 95440 Bayreuth, Germany. ✉e-mail: temesgen.abera@geo.uni-marburg.de

In recent decades, with an increase in the availability of satellite observation data, spatially explicit information on the global extent of deforestation and its impact on local temperature is improving[16–19]. Consequently, a growing body of literature has studied the impacts of deforestation on local temperature using satellite observations, following either a time series[16,18–21] or a space-for-time substitution[17,22,23] approach. These studies are consistent in their findings that deforestation in the tropics leads to greater surface temperature warming (~3 °C maximum) than elsewhere. Nonetheless, the focus has been mainly on radiometric surface temperature[17–19,21], rather than air temperature, and in showing how deforestation impacts vary globally across latitudes. Furthermore, studies using the space-for-time substitution approach have largely excluded montane forests, as this approach is less suitable for areas with large topographic gradients[17,22]. Hence, an exclusive study on the impacts of montane deforestation on CBH and air temperature warming is lacking.

With recent in situ measurements providing evidence of CBH increases in response to canopy removal in tropical montane forests[8], it is crucial to investigate how the impacts of montane deforestation on CBH vary spatially in Africa. In particular, while montane deforestation is accelerating globally (at an annual rate of 0.31%), tropical montane forests in Africa have experienced the highest rate of deforestation in the last two decades (0.48% per year)[24]. Although controversial and geographically variable, local rates of montane deforestation in unprotected areas of Africa can be as high as 3% per year[25]. The main driver of montane deforestation in Africa has been attributed to small-scale cropland expansion, with other factors (e.g., urbanisation, large-scale commodity crops and forest fires) playing a lesser role[26,27].

Here, we investigated the impacts of deforestation in montane forests in Africa over the past two decades with the objective of evaluating and providing 1) spatially explicit information on deforestation-induced maximum air temperature and CBH change, and 2) comparing the relative and net impacts of deforestation-induced and climate change-induced CBH change. The study followed a data-driven approach consisting of three steps (Methods). First, tree cover loss in montane forests, between 2003 and 2022, was identified using the Global Forest Change (GFC) product, and a forest gain filtering procedure, based on a spectral mixture model, was applied to exclude pixels that have recovered after forest loss. Second, air temperature and dew point temperature at a 1-km resolution were estimated by building an ensemble learning model based on satellite observation data, which were trained and validated using 498 weather station data over Africa. The impact of deforestation on air temperature between 2003 and 2022 was estimated using a time series approach after filtering and removing the impact of the concurrent background climate change signal following previous studies[16]. While the deforestation impact on CBH was estimated using an empirical method based on air temperature and dew point temperature[28] (Methods), the climate change impact on CBH was studied using 30 years of data from ERA5-Land for the baseline period from 1992 to 2022.

We show that tree cover loss in tropical montane forests increases air temperature and CBH, yet the magnitude is controlled by the opposing effect between steep elevation gradients and the extent of tree cover loss. A large tree cover fraction loss in a montane forest can offset the cooling effect of elevation gradients. We further show that deforestation has a stronger impact on the magnitude of CBH than climate change does.

## Results

### Widespread forest loss in the lower montane forests across Africa

Between 2003 and 2022, montane forests decreased by 18% with $7.4 \pm 0.5$ million ha of forest lost during this period in Africa (Fig. 1a). When evaluated on a 1 km × 1 km grid, approximately 80% of the forest loss occurred at an elevation ≤1800 m above sea level (a.s.l.), with an average fractional tree cover loss of 15% (Fig. 1b, c). Montane forests that have lost ≥50% of their initial tree cover in 2003 represent only 3.6% of the total forest loss. The number of areas affected by tree cover loss decreases with increasing elevation, whereas the intensity of deforestation (i.e., fraction of tree cover loss) tends to increase with elevation (Fig. 1c, d). Hence, over the last two decades, forest loss in the montane areas of Africa has occurred mainly at lower elevations and in smaller patches.

Forest loss increased the air temperature and CBH to varying degrees across Africa between 2003 and 2022 (Fig. 2). The mean annual average maximum temperature ($T_{max}$) increased by $1.37 \pm 0.58$ °C and the CBH by $236 \pm 87$ m. Locally, $T_{max}$ increased by a maximum of 3 °C and the CBH by a maximum of 300 m, with peaks in both variables occurring between 5 °S and 10 °S latitude. To the contrary, forest loss decreased the dew point temperature by a comparable magnitude of $-1.42 \pm 0.55$ °C (refer to Supplementary Fig. 1 for details of the deforestation-induced dew point temperature change). Additional information regarding the seasonality of deforestation-induced $T_{max}$ and CBH change is provided in Supplementary Fig. 2a, b.

### Elevation and tree loss control magnitude of warming and CBH increase

The influence of elevation and tree cover change on the magnitude of warming and CBH increase is shown in Fig. 3a–c. For comparison purposes, the effect on maximum land surface temperature (LST) is included in the analysis. At lower elevations (1200–1600 m a.s.l.), all variables (LST, $T_{max}$, and CBH) increase with the fraction of tree cover loss until a steep elevation gradient reduces the magnitude of warming and CBH increase despite an increase in the fraction of tree cover loss. However, when the fraction of tree cover loss reaches a certain threshold (i.e., 40% for LST, 44% for $T_{air}$, and 42% for CBH), this trajectory is reversed and the warming and CBH increase steeply regardless of elevation as these thresholds were enough to start offsetting the elevation effect (Fig. 3a–c). The result shows that the contrasting and competing effects between elevation gradient and extent of tree cover loss control the magnitude of warming; locally ~ >70% of the tree cover loss is sufficient to fully offset the cooling effect of elevation and generate as much $T_{max}$ warming as in the lower montane forest.

### Deforestation has stronger impact on CBH increase than climate change

The impact of deforestation on CBH was compared to that of climate change (Fig. 4). Deforestation (2003–2022) had a greater impact on the magnitude of CBH increase than climate change (1992–2022). While deforestation increased CBH across montane forests in Africa, climate change, based on ERA5-Land data from 1992 to 2022, exhibited an increase in 62% and a decrease in 37% of the study area. The net effect was an increase in most (76%) areas of montane forest. In areas where deforestation- and climate change-induced CBH changes are occurring in phase, the CBH increase was magnified. Similar patterns were obtained for $T_{max}$ change (i.e., deforestation increased $T_{max}$ more than climate change) (see Supplementary Fig. 3).

The impacts of deforestation and climate change on CBH at various elevation levels were also evaluated (Fig. 5). The result showed that while elevation reduced the impact of deforestation on CBH ($R^2 = 0.87$, deviance explained = 88%, $P < 0.001$), its effect on the climate change-induced CBH shift was relatively smaller ($R^2 = 0.52$, deviance explained = 53%, $P < 0.001$). At a higher elevation (> ~ 2250 m), deforestation- and climate change-induced CBHs showed an opposite pattern (i.e., with the former decreasing and the latter increasing with elevation).

## Discussion

The present study provides observational evidence on the impacts of deforestation on air temperature and CBH in montane forest in Africa.

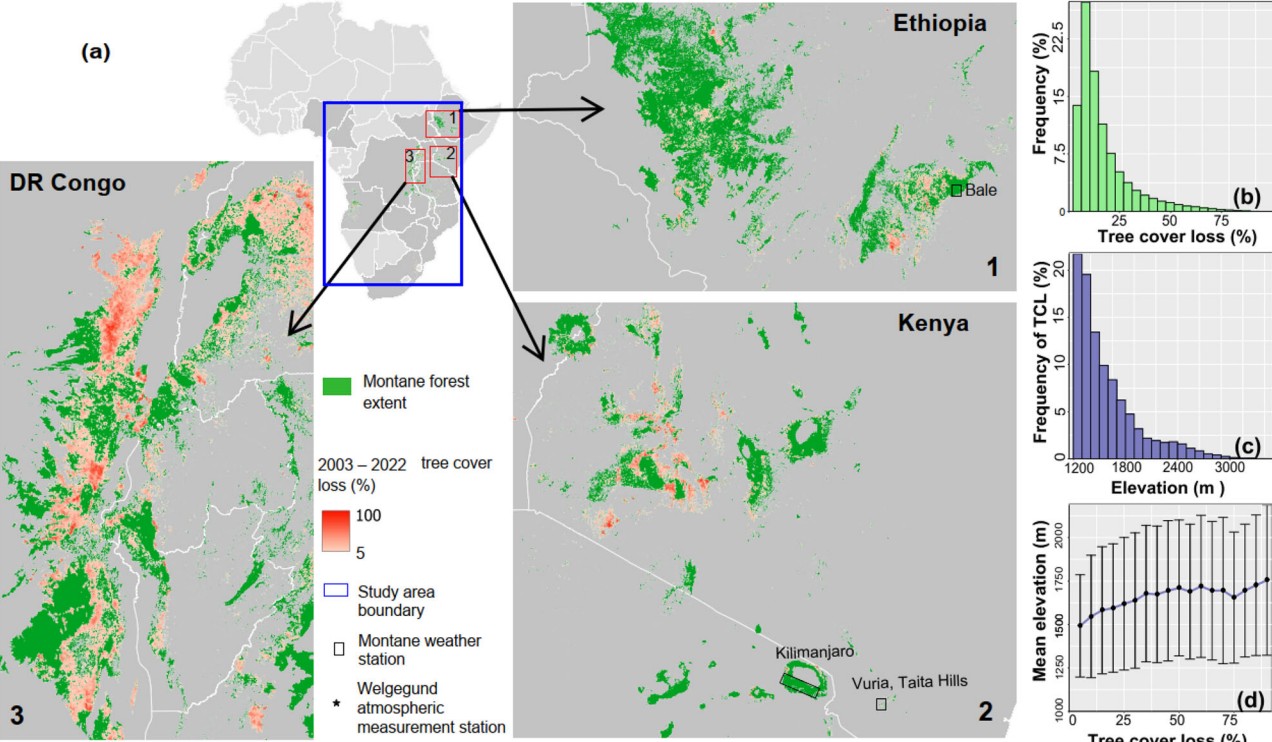

**Fig. 1 | Tree cover loss (TCL) between 2003 and 2022 in montane forests in Africa.** In panel **a** extent of tropical montane forests in Africa (Methods) is used as a background. A closer view of three areas (red boxes) with high concentrations of montane forest is provided. Locations of stations used for external model validation in Bale (Ethiopia), Kilimanjaro (Tanzania), and Welgegund (South Africa) are overlayed on the map. Panel **b** shows the frequency (in %) of TCL. Panel **c** shows the frequency of TCL with increasing elevation in 100-m intervals from 1200 m to 3400 m a.s.l. Panel **d** shows the average elevation for every 5% TCL interval ranging from 5% to 90%. Administrative boundary data are from Global Administrative Areas (GADM)[74].

Our observational deforestation-induced warming result supports previous observational and modelling studies, which reported warming (0.6–2 °C) in tropics[16,29–31]. However, compared to these studies, our result shows a relatively higher warming of up to 3 °C in African montane forests. The differences in the magnitude of warming can be attributed to differences in spatial resolution, extent of study area, length of study period, and methodologies considered. Furthermore, our result showed that the climatic impact of tropical montane deforestation decreases with increasing elevation as the elevation gradient regulates deforestation-induced warming, which agrees with earlier modelling study focusing on tropical mountain regions[29]. The reduction in the warming impact of deforestation with elevation can be explained by the lapse rate (i.e., the decrease in temperature with altitude), frequent cloud cover and immersion, and reduced evapotranspiration with increasing elevation[4,29,32]. However, compared to previous studies, our results indicate that the effect of elevation can be offset by large-scale tree cover loss, and warming comparable to that in lower montane forests can be generated at 70% tree cover loss threshold with spatial resolution of 1 km, providing additional insights into the climatic impacts of montane forest loss. These finding suggests that large-scale deforestation in montane forests is particularly harmful, as its impacts may be greater than previously thought[23].

Compared to previous studies on deforestation-induced cloud regime change in tropical montane forests, our findings provide additional insights. For instance, a previous study showed that lowland deforestation increases the cloud base height in nearby montane forest in Monteverde in Costa Rica[11]. Also, in situ study in the subtropics indicated that the CBH can lift in response to drought and hurricane-induced canopy defoliation in the Luquillo submontane forest in eastern Puerto Rico[8]. Our study extends this understanding to the highlands of Africa, showing that human-induced montane deforestation can increase CBH at the source area of deforestation in Africa. This means that local deforestation in the montane forests deserves attention as it can directly affect the cloud regime lying above the location of deforestation through warming and reduced moisture recycling, as well as in remote areas through teleconnection of moisture transport[6,7,33–35]. This additional information can help to raise awareness of the need to mitigate the impacts of montane deforestation and to conserve and restore of montane forests both inside and outside protected areas.

Our results on climate-induced changes in CBH in African montane forests showed a slightly different pattern than previous global studies[36,37] over the last two decades. For example, both climate models and satellite-based studies reported an average increase in cloud height in response to climate change over the tropics[36,37]. In contrast to these studies, our CBH change showed a latitudinal pattern (i.e., areas south of -17°S latitude showed a decrease in CBH, while those montane forests located north of -17°S latitude, where most of the montane forests in Africa occur, showed an increase in CBH on average). For those areas that showed a decrease in CBH (i.e., south of -17°S latitude), whether the decrease in CBH is related to their geographic proximity to the ocean, differential impacts of climate change, size of montane forest, or other reasons requires further investigation.

## Implications

Deforestation can have far-reaching impacts on the provisioning of montane forest ecosystem services. Montane forests are important sources of fresh water (e.g., for maintaining drinking water, irrigation, and hydropower), and deforestation can threaten the capacity of montane forests to provide fresh water[4,6]. Directly, it alters forest structure (leaf area index, canopy cover, and canopy height) and reduces cloud water interception efficiency (CWI) ([38]; Supplementary

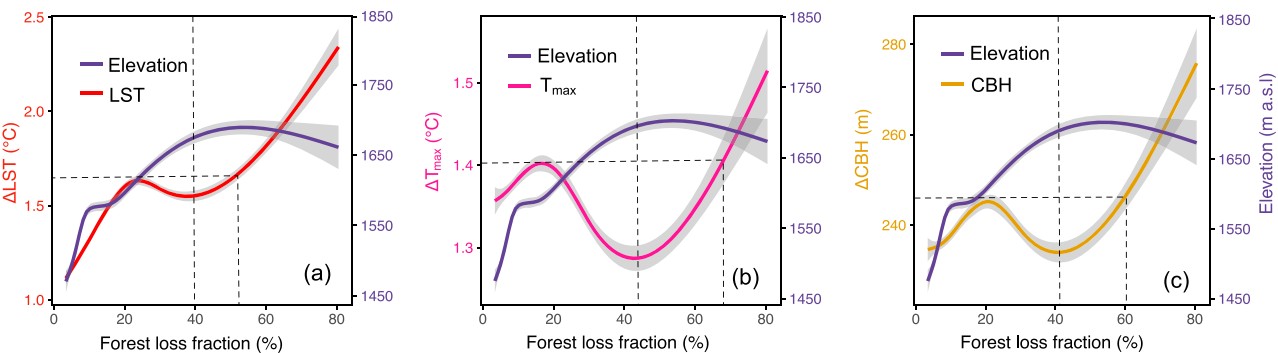

**Fig. 2 | Changes in temperature and cloud base height (CBH) across montane forest in Africa.** Spatial extent and magnitude of deforestation-induced (**a**) maximum air temperature ($T_{max}$) and (**b**) cloud base height change (ΔCBH). Panel **c** shows the deforestation-induced mean ΔCBH and standard deviation (SD) across 1° latitude intervals. Closer views of ΔCBH over Ethiopia (**d**), Kenya (**e**), and central to western parts of eastern Africa (**f**) are displayed for the corresponding boxes in Panel **b**. Tropical montane forests extent in Africa (Methods) is used as a background in panels **d**–**f**. Administrative boundary data are from Global Administrative Areas (GADM)[74].

**Fig. 3 | Relation of fraction of forest cover loss (ΔFCL) with changes in temperature and cloud base height in montane forests in Africa.** Panel **a** deforestation-induced daytime maximum land surface temperature change (ΔLST) with ΔFCL, **b** maximum air temperature change (ΔT$_{max}$) with ΔFCL, and **c** cloud base height change (ΔCBH) with ΔFCL. The line shows a cubic spline regression fit with a 95% confidence interval using a generalized additive model (GAM) for ΔLST vs. FCL ($R^2 = 0.97$; deviance explained = 98%), ΔT$_{max}$ vs. FCL ($R^2 = 0.74$; deviance explained = 77%; $P < 0.001$), and ΔCBH vs. FCL ($R^2 = 0.77$; deviance explained = 80%). Elevation (m a.s.l.) is fitted using GAM for the corresponding FCL in **a**–**c**.

Fig. 4). Deforestation also affects the availability of fresh water supply locally through reduction in evapotranspiration at the source area of tree cover loss and remotely via reducing atmospheric moisture recycling and transport[6,33–35]. Indirectly, deforestation affects CWI by warming the local air temperature[6] and increasing the CBH. An increase in CBH can have a serious implication on freshwater availability in the region. Quantifying the reduction in water supply is particularly important in arid and semi-arid areas, where dry season water availability is heavily dependent on the CWI of montane forests, as it can guide land use interventions such as restoring lost habitat in montane forests due to deforestation and degradation. To what extent rising CBH will affect the availability of freshwater supply in the region

under future land use and climate change projections needs attention and further studies.

In addition to its impact on water supply, deforestation is a major cause of biodiversity loss in the tropics[13,39,40]. Many tropical montane species live in a narrow elevational and climatic range where the montane forests serve as refugia for the survival of plant and animal species[1–4]. However, deforestation-induced warming and CBH-increase can affect biodiversity of montane forest in many aspects[39]. For example, an increase in air temperature alters the microclimate, which is essential for biodiversity conservation through its influence on

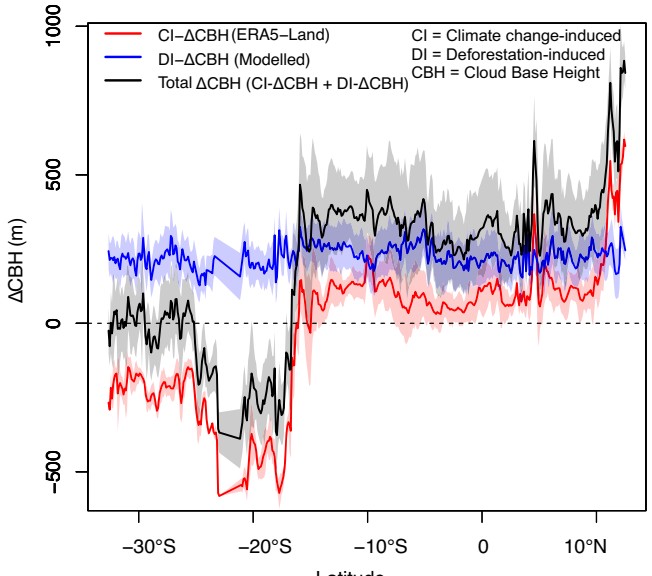

**Fig. 4 | Contribution of climate change and deforestation to mean cloud base height change (ΔCBH) in montane forests of Africa across latitude from South to North at ~10 km interval.** For climate change-induced ΔCBH, 30 years of ERA5-land air temperature and dew point temperature data between 1992 and 2022 were used. Deforestation-induced changes were calculated between 2003 and 2022. The blue, red, and black line show the mean ΔCBH due to deforestation, climate change, and combined effect from these two factors. The shaded region show mean ± standard deviation.

vascular plant species richness, turnover, and composition[41–43]. As the biodiversity of plant and animal species is regulated by thermal tolerance, an increase in air temperature can directly affect the distribution and diversity of organisms and increase their risk of extinction by altering the unique hydro-climatic environment that serves as a micro-refuge for different species[42,43]. Raising CBH also affects biodiversity through its effects on humidity, light conditions, and water availability, which are critical for the survival of organisms[44,45]. For example, an increase in CBH leads to a decrease in humidity, cloud cover, and availability of water from cloud water interception, which are critical for the population size and survival of species[46,47].

Whether montane forest species will adapt to increasing temperature and CBH, migrate upslope, or gradually become extinct remains uncertain. Montane forest species can likely migrate upslope in response to warming and CBH, yet the response may be species specific and some species have limited ability to adapt or move upslope and could become extinct[48,49]. It is also unclear whether species responses to climatic change are location specific and respond similarly or differently across montane forests in Africa and elsewhere in the tropics. A pantropical network of monitoring plots across montane forests would help clarify these questions, and such networks are highly needed[50]. In addition to its impact on biodiversity, montane forests are an important component of the global water, energy, and carbon cycle, and deforestation can further exacerbate global warming through biogeophysical (e.g., reduced evapotranspiration and surface roughness) and biogeochemical mechanisms (e.g., increased CO2 emissions)[16,17,20,51,52].

Compared to climate change, the stronger impact of deforestation on air temperature and CBH calls for more attention to the anthropogenic impact of human activities on montane forests. Because the impacts of deforestation on biodiversity loss may be greater than that of climate change[53], the conservation and management of natural montane forests deserves priority. Furthermore, in most areas where deforestation has increased $T_{air}$ and CBH, climate change also shows a similar increasing pattern. The combined effect of these two factors could amplify the impact on montane forest ecosystems, but whether the impact of climate change will catch up with or exceed that of deforestation under future climate change scenarios is unclear and requires further study.

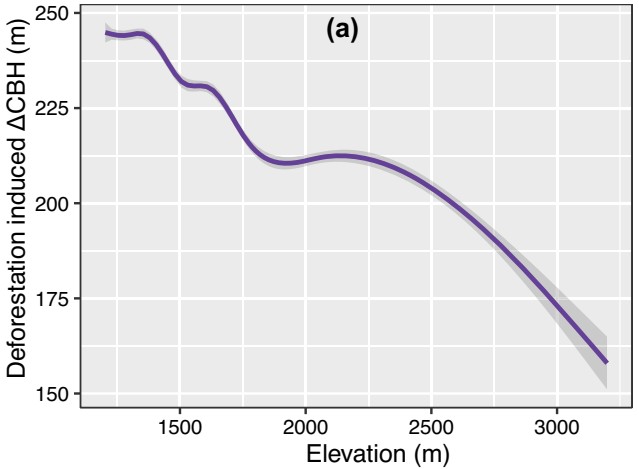
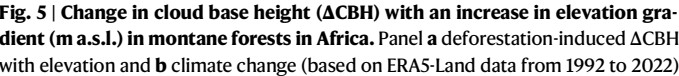
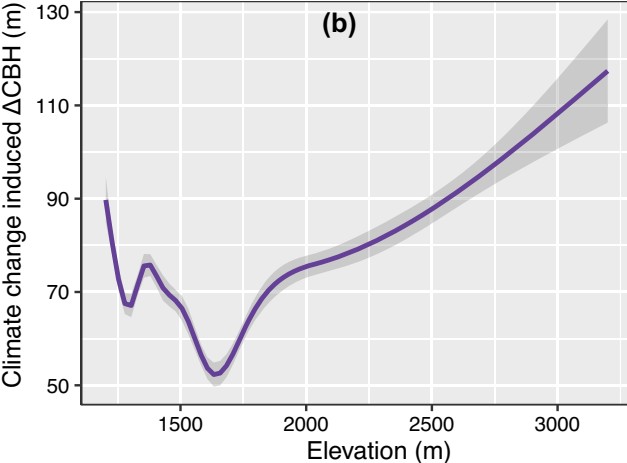

**Fig. 5 | Change in cloud base height (ΔCBH) with an increase in elevation gradient (m a.s.l.) in montane forests in Africa.** Panel **a** deforestation-induced ΔCBH with elevation and **b** climate change (based on ERA5-Land data from 1992 to 2022) induced ΔCBH with elevation. The line shows a cubic spline regression fit with a 95% confidence interval using a generalized additive model.

## Uncertainties and limitations

This study contains uncertainties and limitations related to the data and methods used in our analysis. Previous studies in the tropics reported underestimations of forest loss estimates in the GFC product[54–56]. However, in these studies, GFC product accuracies were not reported separately for highlands and lowlands. Therefore, the accuracy in montane forests across Africa has not been assessed exclusively.

Hence, we evaluated the uncertainty at the 95% confidence interval following sample-based accuracy and area estimation (see Supplementary Fig. 5 for sample locations), as recommended in the literature[57]. Our results showed that the GFC product has an overall accuracy of 95.5% (±1.1%), with an omission error of 18.7% and a false detection error of 7.5%. On the one hand, the omission error indicates that more areas experienced forest loss that were not identified in our estimation, and the impacts of forest loss may be more widespread than what is represented in our result. On the other hand, the smaller false detection error indicates that the uncertainty has a smaller impact on our result.

Our analysis demonstrated local climatic impacts of montane deforestation at a 1 km × 1 km grid. Nevertheless the result does not tell the microclimatic impacts of deforestation, which operate at finer scale than our analysis grid. Exploring the microclimatic impacts of small-scale deforestation is needed in the future. Furthermore, our analysis does not capture the non-local effects of deforestation, such as changes in regional precipitation, cloud cover, and long-wave downwelling radiation[7,58]. Nonetheless, previous modelling and observational studies in the tropics have shown that deforestation reduces cloud cover and observed precipitation[7,58], and thus the non-local effect is likely to amplify the local effects of montane deforestation. On the other hand, deforestation itself can amplify the impact of climate change on CBH through increasing the air temperature via biogeophysical and biogeochemical mechanisms[16,17,52] but these feedbacks were not included in our study.

Our study does not distinguish between plantation forest and natural forest, although doing so would provide better information for forest conservation and management purposes. The lack of recent and spatially complete global or regional products in relation to the extent of plantation and natural forests limits such a classification for the study period (i.e., 2003–2022). Nonetheless, based on the FAO Global Forest Assessment Report for 2020[59], the extent of plantation forest as a proportion of total forest area is very small (i.e., ≤2% averaged over our study area)[59]. Assuming a smaller loss of plantation forest between 2020 and 2022, montane deforestation is likely to have affected natural forests more than plantation forests.

## Methods

### Montane forest extent in Africa

The extent of montane forest was defined based on elevation, tree cover, and local elevation range (LER)[60]. The USGS Digital Elevation model (DEM) with a spatial resolution of 30 m was used to calculate elevation and LER. LER was calculated as the difference between the maximum and minimum elevation within a 5-km radius for each pixel. All pixels with an elevation of 1200–3500 m a.s.l., based on regional groupings of montane forests in Africa[4,61], >300 m LER, and a minimum tree cover of 30%, using the GFC tree cover product for year 2000[62], were considered to be montane forest. The tree cover threshold used here is based on the United Nations Framework Convention on Climate Change (UNFCC) definition of forest (i.e., tree cover of at least 10–30%). Tree cover for year 2000 was processed to discard tree losses that occurred before 2003 to match our analysis period of 2003–2022. The montane forest definition used in this paper does not include subtropical and Mediterranean montane forests, or islands in Africa.

## Identifying deforestation locations between 2003 and 2022

To identify areas affected by tree loss, we used a combined approach based on the GFC product from Landsat time series[62] and a normalized difference fractional index (NDFI) derived from the spectral mixture model applied to Landsat[63], which has been successfully used in previous studies to monitor deforestation in the tropics[64,65]. We used surface reflectance products from Collection 2 Landsat (5,7, and 8) at 30-m resolution to compute the NDFI. To identify stable forest in 2003, all pixels with tree loss recorded between 2001 and 2003 were excluded using the GFC product. If a pixel experienced tree loss after 2003 (i.e., from 2004 to 2022), these pixels were retained. As a pixel can recover from deforestation between 2003 and 2022, a further filtering step was performed using the NDFI trend during this period. This step was necessary because the GFC product only records the presence or absence of tree cover gains between 2000 and 2012. Pixels that showed tree cover gain were discarded through this process. Finally, tree loss pixels identified by the GFC product and confirmed by the NDFI trend (i.e., showing statistically significant tree loss between 2003 and 2022) were used for analysis in this paper.

## Maximum and minimum air temperature modelling

Air and dew point temperature predictions were made using an ensemble machine learning approach (Supplementary Fig. 6). A $T_{max}$ random forest regression model was built using five predictors (land surface temperature, normalized difference vegetation index, albedo, latitude, and longitude) separately for MODIS Terra and Aqua sensors (Eqs. 1 and 2), after performing a spatial forward feature selection using the ffs function in the "CAST" R package. Daytime land surface temperature (LST) data from version 6.1 MOD11A1Terra (10:30 am) and MYD11A1Aqua (1:30 pm) at 1-km resolution and daily temporal frequency were used[66,67], as radiometric thermal emission from the surface directly affects the air temperature. Only good-quality clear sky pixels with LST error ≤1 K were filtered and used in our analysis by applying the flag bits of the quality control layer.

To represent vegetation cover and the fraction of solar radiation reflected or absorbed by a surface, the NDVI calculated from the MODIS Nadir Bidirectional Reflectance Distribution Function (BRDF)-Adjusted Reflectance (NBAR) product (MCD43A4 version 6) and the shortwave broadband albedo product (MCD43A3 version 6.1) at 500-m resolution were used. Latitude and longitude were also used, as air temperature depends on geographic location. All data were spatially and temporally harmonized to 1-km resolution and monthly temporal resolution. The $T_{max}$ model from Terra and Aqua was used as input for the ensemble learning (Eq. (3)). Random forest regression was used both as a base learner in Eqs. (1), (2), (4), and (5) and as a meta-learner in Eqs. (3) and (6) using "randomForest" version 4.7-1.1 R package. $T_{min}$ model was predicted following a similar approach but using night-time LSTs data from Terra (10:30 pm) and Aqua (1:30 am) (Eqs. (4)–(7)). The average air temperature $T_{mean}$ was calculated from $T_{max}$ and $T_{min}$ using Eq. (7).

$$T_{\max(Terra)} = f\left[LST_{day(10:30am),}NDVI,albedo,latitude,longitude\right] \quad (1)$$

$$T_{\max(Aqua)} = f\left[LST_{day(1:30pm),}NDVI,albedo,latitude,longitude\right] \quad (2)$$

$$T_{\max} = f\left[T_{\max(Terra)},T_{\max(Aqua)}\right] \quad (3)$$

$$T_{\min(Terra)} = f\left[LST_{night(10:30pm),}NDVI,albedo,latitude,longitude\right] \quad (4)$$

$$T_{\min(Aqua)} = f\left[LST_{night(1:30am),}NDVI,albedo,latitude,longitude\right] \quad (5)$$

$$T_{\min} = f\left[T_{\min(Terra)}, T_{\min(Aqua)}\right] \tag{6}$$

$$T_{mean} = (T_{\max} + T_{\min}) \div 2 \tag{7}$$

## Dew point temperature modelling

The dew point temperature ($T_{dew}$) was modelled in three steps. First, saturated vapour pressure ($e_s$) was estimated using $T_{\min}$ (Terra) and $T_{\min}$ (Aqua) separately, as $T_{dew}$ is considered to be closer to $T_{\min}$[68] (Eqs.(8) and (9)). Second, $T_{dew}$ was modelled for Terra and Aqua using random forest regression (Eqs. (10) and (11)). Third, $T_{dew}$ (Terra) and $T_{dew}$ (Aqua) were ensembled using random forest to estimate $T_{dew}$ (Eq. (12)).

$$e_{s(Terra)} = 0.6108 \exp \frac{17.27 \times T_{\min(Terra)}}{T_{\min(Terra)} + 237.3} \tag{8}$$

$$e_{s(Aqua)} = 0.6108 \exp \frac{17.27 \times T_{\min(Aqua)}}{T_{\min(Aqua)} + 237.3} \tag{9}$$

$$T_{dew(Terra)} = f\left[e_{s(Terra)}, LST_{night(10:30pm)}, latitude, longitude\right] \tag{10}$$

$$T_{dew(Aqua)} = f\left[e_{s(Aqua)}, LST_{night(1:30am)}, latitude, longitude\right] \tag{11}$$

$$T_{dew} = f\left[T_{dew(Terra)}, T_{dew(Aqua)}\right] \tag{12}$$

## Model validation and spatial autocorrelation test

For model training (70%) and testing (30%), 2-m air and dew point temperature monthly station data for 2003 and 2022 were used (see Supplementary Fig. 7 for the number of samples used for each month, which ranges from 332 to 498 samples) from the National Climate Data Center Global Summary of the Day (GSOD) dataset, National Centers for Environmental Information, National Oceanic and Atmospheric Administration (NOAA), USA[69] (see Supplementary Fig. 7 for GSOD station locations). The GSOD data are available as daily averages, aggregated from hourly station data. We used daily maximum and minimum values. All daily observations were aggregated to a monthly timescale prior to analysis. The model performances for $T_{\max}$, $T_{\min}$, and $T_{dew}$ are presented for each month using root mean square error (RMSE), mean absolute error (MAE), and R-squared ($R^2$) (see Supplementary Figs. 8–10 and Tables 1–3). The presence of spatial autocorrelation in the model was evaluated using spatial cross-validation[70] (Supplementary Fig. 11a–c).

For independent external model validation, air and dew point temperature data from local automatic weather stations in montane forests in Ethiopia (Bale Mountain, Supplementary Fig. 12), Kenya (Taita Hills, Supplementary Fig. 13), and Tanzania (Kilimanjaro Mountain, Supplementary Fig. 14) were used. Data from Ethiopia and Tanzania were obtained from the University of Marburg and data from Kenya from the University of Helsinki. The weather station data, which were collected at 30-minute intervals, were aggregated to a monthly time step to match the temporal scale of our analysis. The validation is presented in Supplementary Fig. 15a–i.

## Estimating deforestation-induced temperature change

The deforestation-induced temperature change (surface, air, or dew point) between 2003 and 2022 was calculated for each pixel using Eqs. (13) and (14). The total temperature change ($\Delta T$) between two years consists of a combined signal from deforestation ($\Delta T_{fcc}$) and climate change ($\Delta T_{cc}$). Hence, the deforestation-induced temperature change ($\Delta T_{fcc}$) was computed by subtracting the climate change signal from the total temperature change using a similar approach in previous studies[16,20].

$$\Delta T = \Delta T_{fcc} + \Delta T_{cc} \tag{13}$$

$$\Delta T_{fcc} = \Delta T - \Delta T_{cc} \tag{14}$$

$\Delta T_{cc}$ was calculated from the temperature change (surface, air, dew point) in nearby stable forests located within 5 km of the tree loss pixels. Stable pixels with no statistically significant change in tree cover and within the elevation range of the deforestation pixels were considered. This was done by first identifying the maximum and minimum elevation within the forest loss pixel and excluding those reference stable pixels outside this elevation range. Furthermore, an inverse distance weighting (IDW) was applied to minimize the influence of distance on the $\Delta T_{cc}$ calculation from stable reference forest pixels (Eq. (15))[16].

$$\Delta T_{cc} = \frac{\sum_{x=1}^{n} \frac{\Delta T_x}{d_x}}{\sum_{x=1}^{n} \frac{1}{d_x}} \tag{15}$$

where $\Delta T_x$(℃) is the temperature change in reference stable forest pixels (x) and $d_x$ is the distance between stable reference forest and tree cover loss pixels (x).

## Estimating deforestation-induced CBH change from 2003 to 2022

The cloud base height (in metres), which is the height at which the atmosphere becomes moisture saturated under adiabatic conditions, was estimated by applying the Epsy equation[28] (Eqs. (16) and (17)). The equation is accurate to within 2% for relative humidity above 50% and temperature between 0 ℃ and 30 ℃[28]. The CBH change ($\Delta CBH$) included a combined signal from forest loss ($\Delta CBH_{fcc}$) and the background climate change signal ($\Delta CBH_{cc}$). $\Delta CBH_{fcc}$ was calculated by removing $\Delta CBH_{cc}$ from our analysis using stable nearby reference forest pixels, following an approach that described the temperature change (Eqs. (18) and (19)). Validation of the CBH estimate against independent in situ CBH measurements in South Africa using a ceilometer is provided in Supplementary Fig. 16 (see also site description).

$$CBH_{(2003)} = 125 \times \left(T_{air(2003)} - T_{dew(2003)}\right) \tag{16}$$

$$CBH_{(2022)} = 125 \times \left(T_{air(2022)} - T_{dew(2022)}\right) \tag{17}$$

$$\Delta CBH = \Delta CBH_{fcc} + \Delta CBH_{cc} \tag{18}$$

$$\Delta CBH_{fcc} = \Delta CBH - \Delta CBH_{cc} \tag{19}$$

where $T_{air}$ and $T_{dew}$ are deforestation-induced air temperature and dew point temperature in 2003 and 2022, respectively. $\Delta CBH_{fcc}$ was used to identify whether the rising temperature due to forest loss in montane forests has lifted the CBH, affecting the ability of montane forests to intercept cloud water, or not.

## Climate change-induced CBH change from 1992 to 2022

The climate change-induced changes in CBH that are unrelated to deforestation operate on a larger spatial scale than deforestation and are estimated using Eqs. (16) and (17) but using 30 years (1992 to 2022)

of air temperature and dew point temperature data from ERA5-Land[71]. In ERA5-Land, vegetation (e.g., leaf area index) does not show inter-annual variation[72]. This static vegetation state makes it preferable for estimating CBH change due to climate change, as it is not affected by tree cover change signals. Selecting ERA5-land over other reanalysis data is due to its relatively high spatial resolution (~10 km) and better accuracy when evaluated against in situ station data in montane forests of Africa (See Supplementary Fig. 17).

We used R 4.3.0 for data analysis and QGIS version 3.30.2 for map composition.

## Data availability

Data used for analysis are available online in the following links. MOD11A1 and MYD11A1 LST product from https://lpdaac.usgs.gov/products/mod11a1v061/ and https://lpdaac.usgs.gov/products/myd11a1v061/, respectively; MCD43A3v061 albedo product from https://lpdaac.usgs.gov/products/mcd43a3v061/; NDVI calculated from NBAR product https://lpdaac.usgs.gov/products/mcd43a4v006/; Landsat 5,7, and 8 surface reflectance https://www.usgs.gov/core-science-systems/nli/landsat/landsat-collection-2-level-2-science-products; 30 m resolution Global Forest Change data from https://glad.earthengine.app/view/global-forest-change; NASA SRTM 30 m DEM from https://glad.earthengine.app/view/global-forest-change; GSOD station data from the National Centers for Environmental Information, NOAA, USA https://www.ncei.noaa.gov/access/metadata/landing-page/bin/iso?id=gov.noaa.ncdc:C00516; administrative boundary data from GADM Version 4.1 https://gadm.org/data.html; data for external validation: in Kilimanjaro mountain (Tanzania) from https://doi.org/10.1594/PANGAEA.942822, in Vuria mountain (Kenya) from Matti Räsänen (email: matti.rasanen@helsinki.fi) at https://doi.org/10.1029/2018GL078837, in Bale Mountain (Ethiopia) from (https://doi.org/10.1016/j.cageo.2020.104641). Data in Welgegund (South Africa) are available from the authors (email: ville.vakkari@fmi.fi and Pieter.VanZyl@nwu.ac.za). The outputs of this study are stored in the public open access data repository at https://doi.org/10.5281/zenodo.12789885 (ref. 73).

## Code availability

All relevant R functions and packages used in this study are referred to in the Methods section. There is no particular code that is considered central to the conclusions.

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

## Acknowledgements

T.A.A. would like to acknowledge postdoctoral funding from the Alexander von Humboldt Foundation (AvH, grant agreement ID: Ref 3.3-1228176-FIN-HFST-P). T.A.A. also thanks the University of Helsinki's Taita Research Station in Kenya and Matti Räsänen for his support in Taita Weather Station data. J.H. and P.K.E.P. would like to acknowledge funding from the European Union DG International Partnerships under DeSIRA programme for ESSA (FOOD/2020/418-132). M.A.M. would like to acknowledge German Research Council (DFG) in the framework of the joint Ethio-European Research Unit 2358.

## Author contributions

T.A.A., J.H., E.E.M., & D.Z. conceptualize the study. T.A.A. designed the methodology and wrote the initial manuscript. T.A.A. & J.H. performed the analysis. D.Z. supervised the study. M.A.M. & N.B. pre-processed weather station data from Ethiopia and Tanzania, respectively. V.V. & P.G.V. collected and pre-processed CBH data from South Africa. D.Z., J.H., E.E., M.A.M., N.B., V.V., B.T.H., P.K.E.P. & A.H. contributed to scientific discussion. All authors reviewed and approved the manuscript.

## Funding

## Competing interests

The authors declare no competing interests.
