## [Peer Review File · Nature Communications]

Deforestation amplifies climate change effects on warming and cloud level rise in African montane forestsREVIEWER COMMENTS

Reviewer #1 (Remarks to the Author):

This is an exceptional submission and deserves to be published as soon as possible. The findings are highly important and need to be made available in the general literature.

I did think that a number of references were perhaps missing, which might help extend some of the analysis in the paper. These, I think, are worth considering in more detail. However, I do not wish to "impose" any of these references on the Authors. They should be considered as potential additional references and possibly discussion points in the framework of a minor revision.

<http://www.sciencedirect.com/science/article/pii/S0921818110001773>

Pepin et al make the argument that the loss of tree cover leads to less ET being deposited on glaciers in the region (Kenya)... See perhaps also Moelg et al (<https://www.degruyter.com/document/doi/10.1525/9780520934245-013/html>)

The following paper by Gerbrehiwot et al (<https://onlinelibrary.wiley.com/doi/abs/10.1002/wat2.1317>) makes the argument that the waters of the Blue Nile Basin depend at least in part on ET from places like the Congo Basin and the West African rainforest. Your submission here really talks about the "receiving end" of this relationship (though certainly not less important). Furthermore, this general relationship is also important with regard to downstream relationships (outcomes) in the Blue Nile Basin and beyond. Some of these points may be worth mentioning here...

It is probably also useful to cite more of the literature on cloud forests and downstream water availability. Thus, for example, you might highlight:

<https://global.oup.com/academic/product/tropical-rain-forest-ecology-diversity-and-conservation-9780199285884>

<http://dx.doi.org/10.1002/hyp.7974>

Finally, I noticed that your T analysis also estimates the impact of albedo... I do have questions about whether or not this is the best possible measure of impacts on T. In this regard, you might consider the findings of the following paper:

<https://onlinelibrary.wiley.com/doi/10.1111/gcb.17195>

The following two references may also be useful:

<http://onlinelibrary.wiley.com/doi/10.1002/joc.3409/abstract>

<https://doi.org/10.1002/joc.3566>

I would like to emphasize that these suggestions are provided in the hopes of helping to broaden and strengthen the discussion in this submission. I think this is generally both an excellent and important paper and look forward to seeing it eventually in published form.

Reviewer #2 (Remarks to the Author):

In summary, this study makes an important advance in quantifying the climatic impacts of montane deforestation in Africa with novel findings that deepen our understanding of how these effects are mediated by elevation and compare to climate change. The results have significant implications for the conservation of these ecosystems and the critical water provisioning services they provide. I recommend publication with some technical clarifications as noted.

-While previous studies have examined montane deforestation impacts in specific locations (mainly in the Neotropics) and often focus on surface temperature, this work is novel and valuable in its Africa-wide scope, use of air temperature which is more ecologically relevant, and consideration of CBH which has important consequences for moisture inputs and cloud immersion. The literature review demonstrates the research addresses an important gap. I am curious to understand the comparison between montane forest deforestation and effects and the northern Andes montane forest, as well as moisture recycling?

-The results that 1) Montane deforestation significantly increased air temperature and CBH in Africa from 2003-2022, with the magnitude mediated by elevation and extent of tree loss, and 2) These climatic impacts of forest loss were greater than those from background climate change over this period or over the 1992-2022 period? I do not understand why they are different in time span.

-The methodology of combining extensive remote sensing data, machine learning models validated with weather stations, and robustness checks gives confidence to the findings, suggestions:

- State the sample sizes and rationale for 70/30 training/test split for the RF models
- Discuss implications of 1-km scale for capturing finer-scale microclimatic gradients in montane forests
- Quantify variable importance and hyperparameter tuning for RF in supplement
- Elaborate how locations and elevations were matched when extracting reference pixels for background climate signal
- List years of station data used for validation to clarify temporal coverage

More details that support reproducibility could be provided in the supplement, but the core methods are sufficiently described for the main text.

Reviewer #3 (Remarks to the Author):

#Review comments

This manuscript delves into an important and timely topic of the understudied role of tropical montane forests have on climate change especially how their loss affects biodiversity, warming and cloud level rise in Africa. It reveals how deforestation is a major cause of biodiversity loss, impacts on Cloud base height (CBH) and warming which is definitely contributing the limited knowledge on the African tropical montane forests. The authors further comprehensively assessed and demonstrated how climate change can impact on CBH and warming. The study provides the essential observational evidence that shows response of air temperature and CBH to deforestation in African montane forests over a period. The study also demonstrates the magnitude of forest loss due to deforestation between 2003 and 2022 and shows that it has led to notable dynamics in maximum air temperature and cloud base height by respective values with various interplay of other factors. The study further indicates that the warming effect of deforestation diminishes with elevation, with climate change effects having the opposite effect. However, tree cover reductions exceeding 70% were enough to counteract the cooling effect of elevation, causing comparable increases in warming and cloud base height even at higher elevations (> 1600 m above sea level).

The methodology adopted by the authors are sound and the remotely sensed data they have assembled is very extensive and sufficient and can be easily reproduced.

I enjoyed reading the manuscript. It is generally well written and structured with minimum errors. The figures are clear and neat with the descriptions and are much self-explanatory. And I feel the Manuscript will be of great value and interest to the readers of this journal.

However, the Manuscript needs to be revised before it can be published.

#Major Revision

The authors need to improve on some of the main (section) text i.e. by showing the necessary statistics on the recent trends of deforestation, especially for the tropical montane forest in Africa and as well at the global scale: there is numerous literature that can offer the authors this information.

The manuscript talks about deforestation affecting biodiversity loss in the tropical montane forests, but do not have any linkages how this was assessed. I would suggest the authors add a section on how biodiversity is affected by deforestation, climate change especially after the findings showing that indeed deforestation and Climate change have an influence on CBH and warming.

Improve on the discussion section by linking their results with the literature and demonstrate how their study has advanced knowledge which is lacking in the tropical montane forest especially in Africa.

Are there similar studies in the rest of the tropical montane forests in the other continents? Or

montane forests out of the tropics? How do they compare with the African tropical montane forests in this current study? This would give a very good context to understand where the tropical montane forests in Africa are.

#Minor Revision

Comments directly placed on the Manuscript PDF file.

Response to reviewers' comment

Reviewer #1 (Remarks to the Author):

This is an exceptional submission and deserves to be published as soon as possible. The findings are highly important and need to be made available in the general literature.

Response #1

We thank the reviewer for the very positive recommendations on our manuscript.

I did think that a number of references were perhaps missing, which might help extend some of the analysis in the paper. These, I think, are worth considering in more detail. However, I do not wish to "impose" any of these references on the Authors. They should be considered as potential additional references and possibly discussion points in the framework of a minor revision.

Response #2

We included all suggested references (e.g., Pepin et al., 2010; Gerbrehiwot et al., 2018; Bruijnzeel et al., 2010; Ellison et al., 2024) among others and added new paragraphs in the Discussion sections of the manuscript to put our results in the context of literatures (e.g., lines 79-84, 201-205, 218-238, and 258-269).

<http://www.sciencedirect.com/science/article/pii/S0921818110001773>

Pepin et al make the argument that the loss of tree cover leads to less ET being deposited on glaciers in the region (Kenya)... See perhaps also Moelg et al (<https://www.degruyter.com/document/doi/10.1525/9780520934245-013/html>)

The following paper by Gerbrehiwot et al (<https://onlinelibrary.wiley.com/doi/abs/10.1002/wat2.1317>) makes the argument that the waters of the Blue Nile Basin depend at least in part on ET from places like the Congo Basin and the West African rainforest. Your submission here really talks about the "receiving end" of this relationship (though certainly not less important). Furthermore, this general relationship is also important with regard to downstream relationships (outcomes) in the Blue Nile Basin and beyond. Some of these points may be worth mentioning here...

It is probably also useful to cite more of the literature on cloud forests and downstream water availability. Thus, for example, you might highlight:

<https://global.oup.com/academic/product/tropical-rain-forest-ecology-diversity-and-conservation-9780199285884>

<http://dx.doi.org/10.1002/hyp.7974>

Finally, I noticed that your T analysis also estimates the impact of albedo... I do have questions about whether or not this is the best possible measure of impacts on T. In this regard, you might consider the findings of the following paper:

<https://onlinelibrary.wiley.com/doi/10.1111/gcb.17195>

The following two references may also be useful:

<http://onlinelibrary.wiley.com/doi/10.1002/joc.3409/abstract>

<https://doi.org/10.1002/joc.3566>

Response #3

We agree with the comment that the albedo impact on T can be smaller specially in low-latitudes (tropics) than high-latitudes. Here we used albedo as it helps to approximate the change in the amount of incoming solar radiation reflected or absorbed during deforestation, which can slightly impact T through its radiative forcing. Our T_{max} model (new section in Supplementary Material “Miscellaneous information related to temperature model”, page 8) further supports that albedo has smaller impact on T as it is the least important variable in predicting T.

I would like to emphasize that these suggestions are provided in the hopes of helping to broaden and strengthen the discussion in this submission. I think this is generally both an excellent and important paper and look forward to seeing it eventually in published form.

Response #4

Thank you.

Reviewer #2 (Remarks to the Author):

In summary, this study makes an important advance in quantifying the climatic impacts of montane deforestation in Africa with novel findings that deepen our understanding of how these effects are mediated by elevation and compare to climate change. The results have significant implications for the conservation of these ecosystems and the critical water provisioning services they provide. I recommend publication with some technical clarifications as noted.

Response #1

We thank the reviewer for the very positive recommendations on our manuscript.

While previous studies have examined montane deforestation impacts in specific locations (mainly in the Neotropics) and often focus on surface temperature, this work is novel and valuable in its Africa-wide scope, use of air temperature which is more ecologically relevant, and consideration of CBH which has important consequences for moisture inputs and cloud immersion. The literature review demonstrates the research addresses an important gap. I am curious to understand the comparison between montane forest deforestation and effects and the northern Andes montane forest, as well as moisture recycling?

Response #2

Regarding effects of montane deforestation in the northern Andes, we speculate that similar effects to the montane deforestation in Africa might happen (i.e., air temperature and CBH can increase in response to deforestation and less moisture can recycling due to reduced evapotranspiration). The montane deforestation in Andes or elsewhere in tropics can affect water availability, not only in the

source area of deforestation, but in remote areas elsewhere through teleconnection of moisture transport and recycling (Espinoza et al., 2020; Pepin et al., 2010). But this speculation requires further study as there can be two-way feedback in the Amazon-Andes hydroclimate interconnection (Espinoza et al., 2020), as moisture inputs from the Amazon to the Andes Mountain might counteract, stabilize, or reduce the deforestation impact in the Andes.

To address the comment, we have included additional sentences in the discussion section (line 245-247) as “Deforestation also affects the availability of fresh water supply locally through reduction in evapotranspiration at the source area of tree cover loss and remotely via reducing atmospheric moisture recycling and transport (34,35,36, 37).”

References

Espinoza et al., 2020. Hydroclimate of the Andes Part I: Main Climatic Features, *Front. Earth Sci.* <https://doi.org/10.3389/feart.2020.00064>

Pepin et al., 2010. The montane circulation on Kilimanjaro, Tanzania and its relevance for the summit ice fields: Comparison of surface mountain climate with equivalent reanalysis parameters. *Glob. Planet. Change* <https://doi.org/10.1016/j.gloplacha.2010.08.001>

-The results that 1) Montane deforestation significantly increased air temperature and CBH in Africa from 2003-2022, with the magnitude mediated by elevation and extent of tree loss, and 2) These climatic impacts of forest loss were greater than those from background climate change over this period or over the 1992-2022 period? I do not understand why they are different in time span.

Response #3

Thank you for the comment. The climate change period refers to 30 years (1992-2022). The reason for using a different time span for the climate change is related to fulfilling the requirements of its definition, which refers to the long-term change in climate over a period of minimum 30 years. Nonetheless, when the same time span was used, the impact of deforestation is stronger in magnitude than what was reported in the manuscript.

To clarify this, we have added time reference in the sentences (line 170 and 171) as “The impact of deforestation on CBH was compared to that of climate change (Figure 4). Deforestation (2003 – 2022) had a greater impact on the magnitude of CBH increase than climate change (1992 – 2022).

-The methodology of combining extensive remote sensing data, machine learning models validated with weather stations, and robustness checks gives confidence to the findings, suggestions:

- State the sample sizes and rationale for 70/30 training/test split for the RF models

Response #4

We now state the sample sizes in the main manuscript (lines 393-394). Details of the sample for each month are also provided in Supplementary Figure 6. The modified text now reads “For model training (70%) and testing (30%), 2-m air and dew point temperature monthly station data for 2003 and 2022 were used (see Supplementary Fig. 6 for the number of samples used for each month, which ranges from 332 to 498 samples).

- Discuss implications of 1-km scale for capturing finer-scale microclimatic gradients in montane forests

Response #5

We included additional information in the limitation section (line 303-306), which now reads “Our analysis demonstrated local climatic impacts of montane deforestation at a 1km x 1 km grid. Nevertheless the result does not tell the microclimatic impacts of deforestation, which operate at finer scale than our analysis grid. Exploring the microclimatic impacts of small-scale deforestation is needed in the future”

- Quantify variable importance and hyperparameter tuning for RF in supplement

Response #6

We now included these in the supplementary material (see new section in Supplementary Material “Miscellaneous information related to temperature model”, page 8).

- Elaborate how locations and elevations were matched when extracting reference pixels for background climate signal

Response #7

We added text for clarification (lines 425-427), which reads “Stable pixels with no statistically significant change in tree cover and within the elevation range of the deforestation pixels were considered. This was done by first identifying the maximum and minimum elevation within the forest loss pixel and excluding those reference stable pixels outside this elevation range”.

- List years of station data used for validation to clarify temporal coverage

More details that support reproducibility could be provided in the supplement, but the core methods are sufficiently described for the main text.

Response #8

For validation (as well as training) we used GSOD monthly data for the year 2003 and 2022 as stated in Supplementary Figure 6. We have now clarified this in the main manuscript as well, which reads “For model training (70%) and testing (30%), 2-m air and dew point temperature monthly station data for 2003 and 2022 were used (see Supplementary Fig. 6 for the number of samples used for each month, which ranges from 332 to 498 samples)” (line 395-396).

In addition, more details are now included both in the supplementary material (see new section in Supplementary Material “Miscellaneous information related to temperature model”, page 8) and Methods (line 353-354, 367, and 467) for reproducibility of our approach.

Reviewer #3 (Remarks to the Author):

#Review comments

This manuscript delves into an important and timely topic of the understudied role of tropical montane forests have on climate change especially how their loss affects biodiversity, warming and cloud level rise in Africa. It reveals how deforestation is a major cause of biodiversity loss, impacts on Cloud base height (CBH) and warming which is definately contributing the limited knowledge on the African tropical montane forests. The authors further comprehensively assessed and demonstrated how climate change can impact on CBH and warming. The study provides the essential observational evidence that shows response of air temperature and CBH to deforestation in African montane forests over a period. The study also demonstrates the magnitude of forest loss due to deforestation between 2003 and 2022 and shows that it has led to notable dynamics in maximum air temperature

and cloud base height by respective values with various interplay of other factors. The study further indicates that the warming effect of deforestation diminishes with elevation, with climate change effects having the opposite effect. However, tree cover reductions exceeding 70% were enough to counteract the cooling effect of elevation, causing comparable increases in warming and cloud base height even at higher elevations (> 1600 m above sea level).

The methodology adopted by the authors are sound and the remotely sensed data they have assembled is very extensive and sufficient and can be easily reproduced.

I enjoyed reading the manuscript. It is generally well written and structured with minimum errors. The figures are clear and neat with the descriptions and are much self-explanatory. And I feel the Manuscript will be of great value and interest to the readers of this journal.

Response #1

We thank the reviewer for the very positive recommendations on our manuscript.

However, the Manuscript needs to be revised before it can be published.

Response #2

We have considered the comments and revised the manuscript as detailed below.

#Major Revision

The authors need to improve on some of the main (section) text i.e. by showing the necessary statistics on the recent trends of deforestation, especially for the tropical montane forest in Africa and as well at the global scale: there is numerous literature that can offer the authors this information.

Response #3

We have accepted the comment and provided background information (including trends of deforestation globally, whole Africa, locally, and its major drivers) from the literature in the Introduction text (line 77-82) as “In particular, while montane deforestation is accelerating globally (at an annual rate of 0.31%), tropical montane forests in Africa have experienced the highest rate of deforestation in the last two decades (0.48% per year) (24). Although controversial and geographically variable, local rates of montane deforestation in unprotected areas of Africa can be as high as 3% per year (25). The main driver of montane deforestation in Africa has been attributed to small-scale cropland expansion, with other factors (e.g., urbanisation, large-scale commodity crops and forest fires) playing a lesser role (26,27).”

The manuscript talks about deforestation affecting biodiversity loss in the tropical montane forests, but do not have any linkages how this was assessed. I would suggest the authors add a section on how biodiversity is affected by deforestation, climate change especially after the findings showing that indeed deforestation and Climate change have an influence on CBH and warming.

Response #4

We included a new paragraph (line 258-269), which reads “Many tropical montane species live in a narrow elevational and climatic range where the montane forests serve as refugia for the survival of

plant and animal species (1,2,3,4). However, deforestation-induced warming and CBH-increase can affect biodiversity of montane forest in many aspects (41). For example, an increase in air temperature alters the microclimate, which is essential for bio-diversity conservation through its influence on vascular plant species richness, turnover, and composition (43,44,45). As the biodiversity of plant and animal species is regulated by thermal tolerance, an increase in air temperature can directly affect the distribution and diversity of organisms and increase their risk of extinction by altering the unique hydro-climatic environment that serves as a micro-refuge for different species (44,45). Raising CBH also affects biodiversity through its effects on humidity, light conditions, and water availability, which are critical for the survival of organisms (46,47). For example, an increase in CBH leads to a decrease in humidity, cloud cover, and availability of water from cloud water interception, which are critical for the population size and survival of species (48,49).

Improve on the discussion section by linking their results with the literature and demonstrate how their study has advanced knowledge which is lacking in the tropical montane forest especially in Africa. Are there similar studies in the rest of the tropical montane forests in the other continents? Or montane forests out of the tropics? How do they compare with the African tropical montane forests in this current study? This would give a very good context to understand where the tropical montane forests in Africa are.

Response #5

We have accepted the comment and added sentences as well as new paragraphs in the discussion section (lines 199-205, 218-238, and 258-269). Copies of the new paragraphs are copied here below.

Lines 218-238

Compared to previous studies on deforestation-induced cloud regime change in tropical montane forests, our findings provide additional insights. For instance, a previous study showed that lowland deforestation increases the cloud base height in nearby montane forest in Monteverde in Costa Rica (11). Also, in situ study in the subtropics indicated that the CBH can lift in response to drought and hurricane-induced canopy defoliation in the Luquillo submontane forest in eastern Puerto Rico (8). Our study extends this understanding to the highlands of Africa, showing that human-induced montane deforestation can increase CBH at the source area of deforestation in Africa. This means that local deforestation in the montane forests deserves attention as it can directly affect the cloud regime lying above the location of deforestation through warming and reduced moisture recycling, as well as in remote areas through teleconnection of moisture transport (7,34,35,36, 37). This additional information can help to raise awareness of the need to mitigate the impacts of montane deforestation and to conserve and restore of montane forests both inside and outside protected areas.

Our results on climate-induced changes in CBH in African montane forests showed a slightly different pattern than previous global studies (38, 39) over the last two decades. For example, both climate models and satellite-based studies reported an average increase in cloud height in response to climate change over the tropics (38,39). In contrast to these studies, our CBH change showed a latitudinal pattern (i.e., areas south of $\sim 17^{\circ}\text{S}$ latitude showed a decrease in CBH, while those montane forests located north of $\sim 17^{\circ}\text{S}$ latitude, where most of the montane forests in Africa occur, showed an increase in CBH on average). For those areas that showed a decrease in CBH (i.e., south of $\sim 17^{\circ}\text{S}$ latitude), whether the decrease in CBH is related to their geographic proximity to the ocean,

differential impacts of climate change, size of montane forest, or other reasons requires further investigation.

Lines 257-267 (see **Response #4** for this paragraph)

#Minor Revision

Comments directly placed on the Manuscript PDF file.

Response #7

We thank the reviewer for the comments. Below, we have provided a point-by-point reply to the comments copied from the annotated PDF in their order of appearance.

The title in the current state limits the significant work the authors have undertaken, it makes it appear as though the authors are ONLY assessing how montane forest loss contributes to climate change effects on warming & CBH yet in reality they have assessed how the loss of tropical montane forests due to deforestation affects CBH & warming. They have also collected extensive data and assessed how climate change affects CBH and warming.

I suggest they authors revise the title to cover the significant work they have undertaken.

Response #8

We have accepted the comment and slightly modified the title to better reflect the content of the study as "Deforestation amplifies climate change effects on warming and cloud level rise in African montane forest".

Reviewer comment on the Main section

Can you specify this nearby forest? say as: ..regime of XXX montane forest in XXX (ref. 11, 14)

Response #9

Done (line 56).

What does the literature say in terms of impacts of deforestation on local temperature. Can you give examples of the figures showing these impacts?

Response #10

Done (line 69-70).

Revise this sentence for clarity so that the focus of impacts of montane deforestation on CBH & warming?

Response #11

Done (line 74-75).

Could you state how high deforestation rates have been is it increasing or decreasing in the past 2 decades?

Response #12

We have clarified the sentence by adding the deforestation rate as well as further details in the Introduction section of the manuscript (see **Response #3** for details).

GFC has been reported to have its limitations such as underestimation and over estimation of forest cover change at local scales. What have the author done to remedy these inconsistencies?

Response #13

One source of error in the GFC product, for forest cover change analysis, is the unavailability of forest gain information after 2012. This means, for example a forest loss pixel identified in 2013 could recover in 2022 and in the absence of forest gain information, the pixel could be considered as forest loss. This increases false detecting error or can cause over estimation of forest loss. To reduce this problem, we used a forest cover change detection algorithm (spectral mixture model) based on normalized difference fractional index (NDFI) as we have described in the methodology section. We used NDFI trend to check whether the forest loss pixels identified by GFC product recovered after deforestation or not. If a pixel recovered its forest cover, we filtered and discarded those pixels. This approach can improve the inconsistency as also indicated in our accuracy assessment (based on reference high-resolution Google imagery and Planet monthly mosaics, Supplementary Fig. 17) that showed smaller false detection error of 7.5% compared with the omission error (18.7%) (refer Uncertainties and limitation section for details; lines 296-302). Furthermore, the relatively good overall accuracy ($95.5\% \pm 1.1\%$) of our forest loss detection result indicates sufficient level of confidence in the approach. However, we believe that still further study needs to be done to reduce the omission error.

Reviewer comment on the result section

Figure 3. Very clear and excellent finding. Could there be an explanation why beyond the various thresholds listed, the trajectory shows a steep increase regardless of elevation?

Response #14

We have clarified the sentence (line 155-156) as “...this trajectory is reversed and the warming and CBH increase steeply regardless of elevation **as these thresholds were enough to start offsetting the elevation effect** (Figure 3a–c).

Figure 4. Could the red, blue and black lines be made thinner?

Response #15

Done.

Could the patterns of deforestation influence climate change - induced CBHs? Can the authors add more explanation on Deforestation/Climate change relationship? The point is, deforestation itself affects climate change and there should be an expectation that where the deforestation rate is high, then it affects climate change.

Response #16

Thank you for the comment. We agree that deforestation itself can affect climate change-induced CBH through its impact on air temperature via biogeophysical and biogeochemical mechanisms. Nonetheless, our study compared the two separately and does not consider these feedbacks as this requires a different study approach (e.g., a coupled land-atmosphere modelling). To address the

comment, we have include additional explanation in the discussion (Uncertainties and limitation section, lines 310-312), which reads “On the other hand, deforestation itself can amplify the impact of climate change on CBH through increasing the air temperature via biogeophysical and biogeochemical mechanisms (15,16,51) but these feedbacks were not included in our study. “

A very clear results! The montane forests have high deforestation happening at the lower elevation as compared to the higher elevation.

Reviewer comment on the discussion section

The authors could expound on this discussion by showing or referring to the specific findings e.g; Their findings Vs what the previous studies have reported

Response #18

We have added new paragraphs in the discussion section (see **Response #4 and #5** for details).

The authors talk about the deforestation being a cause of biodiversity loss yet they focus on its impact on CBH and warming only. It would be worthy to add an explainer linkage between CBHs and Warming on biodiversity.

Response #19

We included these in the discussion section (see **Response #4** for details).

Can the authors link these discussions to some of the specific results, such as where there was a comparison between the impacts of deforestation on CBH and as well of climate change on CBH.?

Response #20

We added new paragraph in the discussion section (see **Response #5** for details).

Can you remove this "in response" or revise the sentence?

Response #21

Done.

As asked earlier, what have authors done to remedy these uncertainties? especially at local scales?

Response #22

Please refer to **Response #13** for details.

Reviewer comment on Method section

How do you harmonise the differences in the definition of forest say GFC Vs UNFCC?

Response #23

To make the two definitions more compatible, we did the following. The GFC forest definition uses trees height threshold of 5m and tree cover is provided from 0 to 100% at 30m spatial resolution (~ 900 m² or 0.09 ha). UNFCC is flexible in its forest definition in terms of tree height (minimum 2-5m), tree cover (minimum 10-30%), and minimum area of land (0.05-1.0 hectares). Hence, to harmonize the GFC to UNFCC forest definition, we used the highest minimum tree cover threshold of ≥ 30% and discarded the rest pixels. The reason behind using 30% tree cover for UNFCC is twofold. The first

reason is related to consistency. Meaning, if we use 10% tree cover, which is the lowest minimum threshold, the corresponding height is 2m, which is not available in GFC product. Using the highest minimum threshold for both height and tree cover improves the consistent of the forest definition. The second reason is accuracy issues (i.e., the use of a higher threshold reduces the risk of including some croplands occurring in the agroforestry in the highlands that can meet the lowest threshold (10% tree cover and 2m height) (e.g., false banana trees in agricultural land in montane forest areas in Taita Hills, Kenya) (see Methods, lines 327-331).

REVIEWERS' COMMENTS

Reviewer #2 (Remarks to the Author):

The authors have thoroughly addressed all my previous comments and made the necessary revisions. Therefore, I recommend the manuscript for publication.

Reviewer #3 (Remarks to the Author):

The authors have addressed the comments clearly and the manuscript can now be published in this current revised form.